# Novel Apparatuses for Incorporating Natural Selection Processes into Origins-of-Life Experiments to Produce Adaptively Evolving Chemical Ecosystems

**DOI:** 10.3390/life12101508

**Published:** 2022-09-28

**Authors:** Robert Root-Bernstein, Adam W. Brown

**Affiliations:** 1Department of Physiology, Michigan State University, East Lansing, MI 48824, USA; 2Department of Art, Art History and Design, Michigan State University, East Lansing, MI 48824, USA

**Keywords:** prebiotic evolution, natural selection, selection, cycles, chemical ecosystems, chemical environments, chemical ecosystems, ultraviolet light, dark, heat, cold, freezing, drying, wetting

## Abstract

Origins-of-life chemical experiments usually aim to produce specific chemical end-products such as amino acids, nucleic acids or sugars. The resulting chemical systems do not evolve or adapt because they lack natural selection processes. We have modified Miller origins-of-life apparatuses to incorporate several natural, prebiotic physicochemical selection factors that can be tested individually or in tandem: freezing-thawing cycles; drying-wetting cycles; ultraviolet light-dark cycles; and catalytic surfaces such as clays or minerals. Each process is already known to drive important origins-of-life chemical reactions such as the production of peptides and synthesis of nucleic acid bases and each can also destroy various reactants and products, resulting selection within the chemical system. No previous apparatus has permitted all of these selection processes to work together. Continuous synthesis and selection of products can be carried out over many months because the apparatuses can be re-gassed. Thus, long-term chemical evolution of chemical ecosystems under various combinations of natural selection may be explored for the first time. We argue that it is time to begin experimenting with the long-term effects of such prebiotic natural selection processes because they may have aided biotic life to emerge by taming the combinatorial chemical explosion that results from unbounded chemical syntheses.

## 1. Introduction

The iconic 1953 Miller experiment [1,2] producing amino acids from methane, hydrogen, ammonia and water vapor subjected to heat and electrical discharges opened the door to the modern era of prebiotic chemical experimentation. In the subsequent seventy years, many variations of the Miller apparatus have been constructed as well as novel designs better adapted to more specific prebiotic chemical syntheses (e.g., [3,4,5,6,7,8]). All previous apparatuses share the common characteristic of being meant mainly or solely to carry out chemical reactions, in most cases aimed at producing specific prebiotic chemical products such as amino acids, nucleic acids, sugars or lipids [9]. However, such chemical experiments fall short of permitting such chemical systems to evolve the spontaneous emergence of adaptive chemical ecosystems because the apparatuses used to carry them out do not incorporate two key processes required of evolutionary systems.

According to Darwinian evolutionary theory, four processes are at work in biologically evolving systems that presumably need to be present in analogous forms in evolving chemical ecosystems: (1) individuals within a species vary and species themselves differ from one another in the degree to which they are adapted (that is to say, survive relative to each other) in their environment; (2) individual and species variations are reproducible to the extent that some members of each generation resemble their parents more than other members; (3) natural processes act non-randomly to select among the reproducible variants, altering the rates at which variants survive and reproduce; (4) the most highly reproduced variants form the populations from which further variations can arise and be selected [10,11]. In this manner, biological entities continuously evolve to adapt to their changing environments. Of these four processes, prebiotic chemical experiments have satisfied only the first and second processes. The first process is satisfied by the ability to produce a wide range of chemical species that include biotically relevant ones. The reproducibility criterion can arguably be said to be met because the reactants in each of these chemical processes reliably generate these biotically relevant products so that there is a ready (often steady-state) supply within that chemical environment of the relevant species [12]. Such a continuous resupply of product can be equated with reproduction in biological systems, even though the mechanisms differ.

While the production of a range of prebiotic compounds is a necessary precursor to the evolution of adaptive chemical ecosystems, the greater the number of starting compounds and the longer reactions are run, the greater the number of types of compounds that result leading to what has been called the “prebiotic combinatorial chemical explosion” [13]. There are only three ways to control such combinatorial explosion and those are to selectively eliminate some of the compounds [13]); preferentially produce specific compounds through the emergence of stable synthetic or reproductive cycles [14,15]; or to stabilize and preserve selected compounds by the formation of molecular complexes [16,17]. We argue that it is time to begin experimenting with the long-term effects of the prebiotic natural selection processes that may have aided biotic life to emerge by taming the combinatorial chemical explosion. Little research, especially experimental, has been devoted to this problem [13,18,19,20]. Among the challenges has been the fact that most synthetic experiments have been limited to a week or less in duration [21] so that apparatuses that can explore longer-term experiments are needed such as that described, for example, by Asche et al. [22]. These new apparatuses need to make possible the evolution of chemical ecosystems through adaptation to selection pressures acting over long enough periods of time so that only some compounds “survive” and the selected chemical “species” can be subjected to further variation and adaptation.

A number of such natural selection processes would have been present during the origins of life, including light-dark cycles, freeze–thaw cycles, high heat-cool cycles, wet-dry cycles, and the presence of catalytic surfaces such as clays or minerals, sometimes working in various combinations. While each of these processes have been used previously to aid particular types of prebiotic chemical synthesis experiments (see below), few have been explored as means of prebiotic natural selection and, as far as we are aware, their combined effects have hardly been explored. In order to facilitate such selection experiments, we have prototyped two types of apparatuses that can be run continuously over long periods of time (months and perhaps years) and that incorporate all these selection processes and permit many of them to be run concurrently or serially.

The choice of prebiotic natural selection processes to incorporate into our apparatuses was made through consideration of physical factors present in prebiotic environments that could potentially act as selection agents on chemicals produced by Miller-like experiments. The following provides a brief review of these physical factors, which include cycles of wetness/dryness, heat/cold, light/dark, and presence/absence of potentially catalytic surfaces. We did not consider chemical factors (e.g., the types of gases or compounds, their purity, etc.) in the design of the apparatuses, although it is obvious that these would affect the outcomes of experiments because the choice of gases did not affect apparatus designs. We did, however, design the apparatuses to be re-gassed and the mineral content of the water supply replenished so that a constant resupply of chemical reactants was available to drive production of products (“species reproducibility”). A very brief, representative review of some ways in which each of potential selection factors has been used previously in prebiotic reaction processes follows, which is the result of a broad but not exhaustive literature search for studies using the four processes chosen above. This overview established the general nature of current uses of each process to drive prebiotic reactions, which was followed by a more intensive search on PubMed, and Scholar Google, using appropriate key words, to determine whether special apparatuses have been designed to carry out such processes and whether any are capable of long-term or automated use. Finally, a similar search was made for use of these processes for natural selection of prebiotic compounds. We make no claim of completeness and apologize if we have overlooked any obviously relevant sources.

### 1.1. Wet/Dry Cycles

Wet/dry cycles could have occurred in a number of ways under prebiotic conditions, ranging from daily or weekly cycles in tidal pools, ponds or puddles, to fluctuating hydrothermal pools, to seasonal cycles such as the monsoons that cause periodic flooding in some desert regions [23]. No studies using wet/dry cycles to select out prebiotic products or to evaluate destruction/disappearance of such products during cycling were located. Wet/dry cycles have been employed to drive a number of types of prebiotic reactions such as the polymerization of peptides from amino acids [24,25,26,27], ribonucleic acid (RNA) polymers from ribonucleotide precursors [23,28,29], and to drive the encapsulation of such polymers in lipid protocells [30,31,32]. Fox et al. [4] have created an automated, autonomous apparatus to create such wet-dry cycles called the ‘wet-dry apparatus’ (WDA), which permits variable-length cycles and temperatures during the cycle phases. It is capable of operating in an oxygen-free (pure nitrogen) atmosphere but not, apparently using the kind of ammonia-methane-hydrogen mixtures employed in Miller-type experiments. In fact, it appears that no one has subjected Miller-type product mixtures to such wet/dry cycles either to drive polymerization and lipid aggregation reactions or to study whether such cycles deplete some products while benefiting the survival of others.

### 1.2. Freeze/Thaw Cycles

Freeze/thaw cycles are a normal part of seasonal, and sometimes daily, temperature changes in much of the Northern and Southern hemispheres of the modern-day Earth, in mountainous regions even near the tropics, and occurs extraterrestrially on Mars, various planetary moons of the Solar System and may be of relevance to understanding prebiotic chemistry on comets. Because of the Earth’s rotation and tilt, such cycles may have begun soon after the Earth cooled sufficiently to support prebiotic chemistry some 4 billion years ago or more [33].

Freeze/thaw cycles are often used in prebiotic chemistry experiments for promoting chemical syntheses. The basic premise of such experiments is that as a solution reaches its freezing point, solutes form pockets of highly concentrated eutectic solutions favoring some condensation and polymerization reactions. Lowering the temperature further may then protect the products thus formed. Using this approach, purine and pyrimidines have been synthesized from hydrogen cyanide or cyanoacetylene reactants [34]; ligation of nucleic acids into polymers [35]; RNA copying [36]; ribozyme assembly [37]; selection for optimized ribozyme function at freezing temperatures [38,39]; and the encapsulation of RNA within lipid membranes [40]. Notably absent from the literature are reports of the successful use of freeze/thaw cycles to polymerize peptides from amino acids or complex sugars from monomers, though whether this absence is due to the failure of this method for these compounds or lack of relevant experimentation is not clear.

Freeze/thaw cycles can also act as selection processes, although little emphasis has yet been put on these effect in prebiotic chemical experiments and most of what is known comes from studies from the food industry, analytical chemistry, biochemistry, and medicine. Cycles of freezing and thawing quickly degrade some amino acids (especially beta-alanine, cysteine, glutamic acid and aspartic acid) while others, such as valine and leucine, are not affected [41]. Some peptide hormones such as adrenocorticotrophin, are rapidly degraded as a result of repeated freeze–thaw cycles, but others, such as thyrotropin-releasing hormone, are not [42] and as little as 24 h at 40 °C causes some proteins to degrade into constituent amino acids and other metabolites, a phenomenon enhanced by repeated freeze–thaw cycles [43]. On the other hand, the presence of some amino acids, such as glutamate and lysine, can protect peptides and proteins from freeze–thaw related degradation [44], so that small-molecule interactions need to be considered in complex prebiotic systems. Freeze–thaw cycles also promote cross-linking of proteins, especially those with oxidizable side chains such as cysteines [45]. Furthermore, freeze–thaw cycles preferentially degrade some RNA [46] and DNA [47] sequences in preference to others.

In sum, freeze/thaw cycles have clear relevance to prebiotic chemical conditions but have hardly been investigated in that context and no specialized apparatus to perform prebiotic freeze/thaw experiments appears to have been described.

### 1.3. Ultraviolet Light/Dark Cycles

Light-driven chemical reactions are a common aspect of life on Earth and light of various wavelengths, especially in the ultraviolet (UV) range (generally defined as wavelengths between 10 and 400 nm) has often been employed in prebiotic chemistry experiments since absorption of UV light can enhance the chemical reactivity of some classes of compounds. It is generally assumed that the amount of UV light reaching the prebiotic Earth was significantly greater than it is today because UV light is efficiently absorbed by ozone, which was essentially absent from the atmosphere until about 600 million years ago [48]. UV light is also absorbed below 204 nm by carbon dioxide (CO_2_) and below 168 nm by water or water vapor [48]. The relative absence of UV absorbers on Mars and many other planets and in outer space also makes UV light a potential factor in driving prebiotic reactions in these environments. A more contentious question is whether bodies of water absorb sufficient UV light to prevent it from driving prebiotic chemical reactions or acting as a selection factor during the emergence of life. Experiments show that UV absorption is highly dependent on the solutes contained in any particular body of water, prebiotic freshwater sources being largely transparent to UV while salt waters specifically decrease shortwave (≤220 nm) UV flux and iron-rich waters can be UV-opaque [49]. Thus, salt and iron-rich waters may protect prebiotic organic compounds from degradation by UV light [50]. Overall, the effects of UV light on prebiotic chemistry may vary by geography and hydrogeology and, in the cases in which UV light is combined with wet/dry and freeze/thaw cycles, by the stage of the cycle at which it is present or absent.

Notably, “UV light… may have both constructive and destructive effects for prebiotic syntheses” [49] and is thus another good candidate for acting as a selection factor (both positive and negative) on the products of prebiotic syntheses. Synthetically, UV light acting on simple mixtures of methanol-ammonia-water or acetone-ammonia-water yielded fundamental chemical precursors to many more complex molecules such as methylisourea and acetamide [51] and has driven a variety of prebiotic reactions involving hydrogen cyanide (HCN), sulfites, and sulfides [3]. UV light has also been used to drive the production of adenosine from its precursors and the synthesis of ATP from a combination of adenine and ribose [52,53]. UV-light-driven syntheses of nucleic acid bases, nucleosides, sugars, and ribonucleotides from reactants as simple as formamide and urea have been achieved with appropriate mineral catalysts such as titanium dioxide [54,55,56] and the UV light also has the benefit of destroying many of the side-products of the desired reaction [55].

UV light can also drive the synthesis of [2Fe-2S] and [4Fe-4S] clusters through the photooxidation of ferrous ions and the photolysis of organic thiols [57]. These iron-sulfur clusters can coordinate to and be stabilized by cysteine-containing peptides and mediate the assembly of iron-sulfur cluster-peptide complexes that can drive enzymatic reactions. Iron complexes have also been implicated in the production and breakdown of universal metabolic precursor compounds [58].

Less research seems to have gone into UV-driven synthesis of amino acids and peptides because many of these tend to be susceptible to UV degradation but Ponnamperuma and Peterson [59] reported UV-induced peptide formation from amino acids in the presence of cyanamide. More commonly, UV light tends to break down amino acids that can absorb it. In amino acid mixtures, UV light destroys tyrosine, tryptophan and cystine [60,61] whereas glycine and proline are quite stable except in the presence of salts, oxychlorines or high humidity [61]. Arginine is also stable to UV light but breaks down mainly into urea and ammonia in the presence of a combination of peroxides and UV light [62]. However, some of the breakdown products of amino acids are other amino acids: arginine exposed to UV light (254 nm) in the presence of hydrogen peroxide was partially converted to ornithine, norvaline, serine and aspartic acid [62] while UV light (200–400 nm) not only degraded phenylalanine, tyrosine and tryptophan into smaller fragments but converted a small proportion of each into the others [63,64,65]. Moreover, the breakdown products of these compounds tended to progressively protect the original molecule over time [65]. Thus, the effects of UV light on amino acids and their peptide and protein formation are complex and dependent on the presence or absence of oxidizing agents as well as other solutes.

Most UV light sources that are commercially available have limited spectral ranges but Rimmer et al. [3] have devised an apparatus specifically to simulate the range and intensity of actual solar UV radiation and they have explored the amount of UV exposure required to drive various prebiotic reactions.

### 1.4. Catalytic Surfaces

Catalytic surfaces including clays, minerals, rocks, glasses and even meteoritic dust, have also been explored as means of increasing rates of synthesis and polymerization of prebiotic molecules including amino acids, nucleic acids, peptides and polynucleic acids [12,54,55,56,66,67,68,69,70,71,72,73]. The literature concerning the use of such natural catalytic surfaces is so large that no attempt is made here to encompass its range. Notably, however, we found no references to the use of such materials as means of selecting among prebiotic molecules or their reaction pathways, although this would seem to be an area of possible importance. Additionally, no specific apparatuses appear to have been reported for the implementation of such catalytic surfaces, probably because these can easily be incorporated into existing Miller-style apparatuses.

### 1.5. Combining Selection Processes

In many prebiotic environments, multiple selection factors would have been at work on the chemical ecosystem, yet combinations of the four selection factors described above have rarely been explored. A few representative examples follow.

As noted above [54,55,56], titanium dioxide has been used as a catalyst to promote UV-light synthesis of nucleic aicds. Campbell et al. [74] have employed deliquescent minerals to modify and improve the effectiveness of dry-wet cycling in the production of prebiotic condensation reactions. Combinations of such deliquescent minerals are likely to be even more active because they exhibit increased water adsorption at lower relative humidities resulting in solid dissolution and an increase in chemical reactivity [75]. Shankar et al., [76] and Baú et al. [77] found that peptide polymerization was increased during wet-dry cycling when performed in the presence of goethite (Fe_2_O_3_). Ice has also been explored as a surface upon which UV light can act to synthesize amino acids, quinones and amphiphiles [78,79] and to promote the conversion of thymine into the other nucleobases [78]. Such conditions might have occurred both in space and in the polar regions of planets such as Earth and Mars. Finally, Lin et al. [80] have demonstrated that adsorption of peptides onto montmorillonite protects them from UV irradiation thus raising the possibility that combinations of these physicochemical processes act in competing ways that enhance the survival of some prebiotic species at the expense of others. Many other combinations are clearly possible but very few seem to have been tested.

Once again, no special apparatuses appear to have been devised for these combined process experiments.

### 1.6. Thermal Vents

One type of process that our apparatuses do not incorporate are those involved in thermal vent reactions. Alkaline hydrothermal systems containing catalytic minerals such as Fe(Ni) in the presence of sea water and vent gases such as S_2_, H_2_ and CO_2_ might have driven organic syntheses capable of producing the range of prebiotic compounds necessary to drive the evolution of metabolic and genetic processes [81,82]. Several apparatuses for modeling such hydrothermal vent reactions already exist [83,84,85]. Because these reactors model processes that are largely isolated from the others described above and are difficult to incorporate into a single apparatus of the type we were designing, we felt that it was justifiable not to try to incorporate their features into our novel apparatuses.

### 1.7. Introduction Summary

As the preceding overview demonstrates, a range of physicochemical processes including wet/dry cycles, freeze/thaw cycles, UV light/dark cycles and the use of catalytic surfaces and materials, have been used to increase yields of prebiotic compounds. However, only rarely have such processes been used in prebiotic experiments as natural selection factors. A clear need therefore exists for apparatuses capable of modeling not only individual types of physicochemical selection process cycles but to combine them over extended periods of time so that their effects on complex chemical environments can be analyzed in terms of the evolution of the balances between both synthesis and selection acting simultaneously.

## 2. Materials and Methods

We have invented two apparatuses designed to combine the multiple physicochemical processes listed above either simultaneously or in alternating cycles over extended periods of time (weeks/months).

The first apparatus, called “ReBioGeneSys 1.0” deconstructed Miller-style apparatuses to create a series of connected mini-environments each capable of carrying out a single physicochemical process and connected in such a way that exposure to any particular environment can be carried out in any order desired. The second apparatus, called “ReBioGeneSys 2.0” reintegrates the various processes into a much simpler configuration that once again resembles a single environment exposed to multiple selection processes.

### 2.1. ReBioGeneSys 1.0

ReBioGeneSys 1.0 consists of six primary components along with an variety of hardware to connect and mount these components: (1) gas cylinders, a vacuum pump, and a manifold for controlling their connection to the rest of the apparatus; (2) five 5 L flasks with connections to the gas/vacuum manifold and with connections between the flasks for moving liquid between them; (3) an array of peristaltic pumps for moving liquid between the five flasks; (4) a bespoke, programmable electronic control system for controlling the peristaltic pumps; (5) a specially designed shelf for holding the flasks and their connectors; (6) a Marx generator for reliably producing high voltage sparks at frequent (5 to 10 s) intervals (Figure 1 and Table 1).

The gas/vacuum manifold (Figure 2) essentially consists of a series of Swagelok valves connected in such a way as to permit all of the gas cylinders to access all five of the flasks via a single line so that the flasks share a common atmosphere at a common pressure. The vacuum pump is also connected through the manifold via a valve that shuts off all of the gases permitting all of the flasks to be evacuated simultaneously and any liquids in them to be degassed. The manifold also permits the apparatus to be regularly regassed so that the synthetic conditions are maintained at a relatively constant concentration, thereby avoiding the need to seed new containers at each step of a repeated synthetic process as is done in some current experiments (e.g., [12,13]).

Each five liter flask has a unique function and was modified specifically for it. All flasks are connected to the gas/vacuum line via Swagelock connectors into threaded plastic (PFA) plugs (Figure 3). The central flask has four inlets in its sides and four outlets in the bottom. The four inlets permit fluids to be pumped in from the outlet ports of the other four flasks via Tygon tubing while the four outlets permit the fluid in the central flask to be pumped into the inlets via Tygon tubing in any of the other four flasks. In addition to a single inlet and outlet port, each of the four other flasks has its own function. One has been modified to incorporate a condenser coil through which antifreeze solution is pumped by a chiller/circulator with variable temperature control (Figure 3). The condenser can be used to freeze and thaw the liquid in the flask by varying the temperature appropriately. It can also act as a heat sink for the system if the heating mantle is used to increase the temperature in the next flask, which can be varied from room temperature to boiling thus permitting this flask to either be evaporated via the tubing into the other flasks or to act as a model of a hot spring. Another flask has an insert into which a light source such as a UV lamp can be placed to irradiate the liquid in the flask (Figure 3), modeling light/dark cycles. Furthermore, the final flask has been modified to accept a pair of electrodes that are attached to the Marx generator, permitting the atmosphere to be subjected to rapid, periodic, very high voltage electrical discharges for extended periods of time.

Liquid flow between the flasks is controlled by a set of eight peristaltic pumps (four to move liquid into the central flask from any of the others and four to move liquid from the central flask to any one of the others) via Viton tubing. The pumps are mounted in a Delrin case (Figure 4) and controlled by a programmable electronic switch box (Figure 5). The organization of the pump/tubing connections to the flasks is illustrated in Figure 5.

The Marx generator supplying electrical discharges to the apparatus (far right flask in Figure 1 and Figure 5) is based on well-known principles and schematics and consists essentially of a series of ten capacitors that each accumulate a charge permitting the voltage to be increased from about 9 volts direct current to about 250,000 volts every five to ten seconds (Figure 6 and Figure 7).

### 2.2. ReBioGeneSys 2.0

While ReBioGeneSys 1.0 permits automation and a great amount of experimental flexibility in terms of combining selection processes in varying orders for varying amounts of time, we recognized that a simpler and less expensive apparatus might also have benefits. ReBioGeneSys 2.0. integrates the five flasks of ReBioGeneSys 1.0 into a single apparatus with three glass-connected flasks performing the same range of functions (Figure 8). This configuration brings the apparatus back to something resembling the original Urey-Miller apparatus but with additional functions integrated into the modified design.

The essence of the redesign is begin with the modified Urey-Miller apparatus that we hope to report elsewhere and to integrate an additional flask between the one housing the electrodes and the one that is heated. This additional flask incorporates an opening into which a UV light can be inserted, sealed and clamped in place, as well as a condenser coil that can act as a heat sink or means of freezing and thawing the liquid condensing in the flask. In order to accumulate this liquid, a valve has been inserted below this flask attached to it by means of Masterflex ^TM^ I/P ^TM^ Precision C-Flex tubing. This is special, very thick tubing that resists collapsing during the high vacuum conditions used to evacuate the apparatus and to de-gas liquids added to it, as well as being highly resistant to any ammonia that may be in the experimental atmosphere used during experimentation. Additional details concerning unique components of the apparatus are provided in Figure 9 but, in general, the components do not differ significantly from those already in use in many other Urey-Miller type apparatuses or those described above for ReBioGeneSys 1.0.

## 3. Results

### 3.1. Tests of the Apparatuses

Unlike apparatuses designed to mimic one particular environment, which can be tested by determining whether the apparatus functions to the specifications of that chosen environment by producing specific results under tightly determined conditions (e.g., [1,2,82,83,84,85], the apparatuses described here have no specific function but are designed to be extremely flexible and have a large number of possible experimental permutations. Tests of the apparatuses therefore require comparisons between the results obtained using one selection pressure (say wet/dry) versus another (freeze/thaw) under otherwise strictly comparable conditions. We have not yet performed such experiments and cannot, therefore, validate the utility of the apparatuses in comparison with any existing set of data.

However, we have performed long-term fundamental tests of robustness: both apparatuses are capable of performing simple Miller-type experiments in which each component is activated serially. Both apparatuses can hold a vacuum for extended periods of time (days); liquids move through the apparatuses as designed; all the individual elements (heating, freezing, electrical discharging, etc.) function over extended periods of time; the electrical control system for ReBioGeneSys 1.0 performs as designed, moving the liquids from one flask to another in any programmed order; the freezing elements are robust enough to perform for many weeks through freezing and thawing cycles without cracking; the Marx generator design is robust enough to perform for many months or years (which we know because we have used the same Marx generators in previous Miller-like designs (unpublished data).

### 3.2. Limitations of the Apparatuses

One important limitation which we have found is that the Marx generator needs to be shielded from other electrical elements of the apparatus such as the controller for the pumps and the chiller, using a Faraday cage-like wire mesh. Failure to turn off an unshielded Marx generator while attempting to initiate pumping in ReBioGeneSys 1.0 can short out the electronic controller while failure to turn it off an unshielded Marx generator in both apparatuses can result in interruption of the chiller/circulator function.

The apparatuses have several other experimental limitations as well. They cannot accurately reproduce the conditions that would be found at temperature extremes such as well below the freezing point of water or above its boiling point. They cannot accurately reproduce phenomena such as undersea vents. Their efficiency and effectiveness are very difficult to test completely because they permit a very large number of process permutations, some of which may turn out to interfere with each other or to cause apparatus failures that were not foreseen. If ammonia is employed as one of the gases, the rubber sampling ports degrade of a period of a couple of months and need to be replaced, which may interrupt long term experiments. Furthermore, the borosilicate glass begins to degrade after several years of use (unpublished data) eluting boron compounds into the reaction vessel. Thus, the glass components may need to be replaced regularly as well.

An additional limitation of the current apparatuses is that they are not robotized nor connected directly to chemical detection equipment such as gas chromatography-mass spectrometry as Asche et al. [22] have done with their long-term prebiotic apparatus. Thus, additional innovations that improve performance are undoubtedly possible and hopefully these apparatuses will stimulate such innovations.

## 4. Discussion

The development of apparatuses that can run for extended periods of time and that incorporate physicochemical processes capable of naturally selecting among the diverse products of prebiotic chemical experiments is a necessity if origins of life research is to move beyond chemical syntheses to prebiotic evolutionary systems. As Brunk and Marshall [86] have argued, the next phase of origins-of-life experimentation needs explore more complex chemical ecosystems over longer periods of time under non-equilibrium conditions involving “containment, steady energy and material flows, and structured spatial heterogeneity from the outset”. The apparatuses described here are capable of performing these necessary functions and ReBioGeneSys 2.0 in particular is based on a simpler, Urey-Miller-like apparatus that has been extensively tested over several years to run multi-week and multi-month prebiotic syntheses (unpublished data). The novelty of these new apparatuses is to incorporate for the first time wet/dry, freeze/thaw, and UV light/dark cycles while also permitting simultaneous or separate use of catalytic materials such as clays or minerals. These options open a wide range of novel possibilities for new types of long-term experiments in the evolution of prebiotic chemical ecosystems under varying selection pressures. Such experiments would help to fulfill the suggestion by Stüeken et al. [87] that future prebiotic evolution experiments try to mimic the interaction of multiple environments that may have been present in the Hadean Earth and explore the interactions of their various products.

Some aspects of the apparatuses could undoubtedly be improved in order to optimize their adoption by scientists experimenting with prebiotic systems. One of the major limitations of the two systems built thus far is the large scale and the size makes them difficult to install, expensive to build, and potentially dangerous because of the large amounts of gases employed. While ReBioGeneSys 2.0 attempts to remedy some of these problems, it continues to fall short of one of the intended goals and that is to be an open source platform to enable many researchers to explore many new types of experiments. To remedy this the authors have been steadily working on a miniaturized, microfluidic version of ReBioGeneSys. The intent of the new Mini-ReBioGeneSys is to producible it in multiples and small enough to easily fit in a chemical hood or even in one’s hand.

One option is miniaturization of the current designs, which is clearly possible without great difficulties for both types of apparatus. However, some limitations are inherent in the materials, such as the fragility of glass as it becomes smaller and thinner, the need for the glass to support Swagelok connectors, tubing and compression fittings to control gassing of the apparatus, and the lack of availability of these below 2-to-3 mm in diameter (https://www.swagelok.com accessed on 17 September 2022). There are also increasing challenges making bespoke glass apparatuses of ever smaller scales that can incorporate sampling ports, electrodes, chiller coils, etc. Sleeve stopper septa, which are used for sampling ReBioGeneSys 2.0 do not appear to be available below 2.4 mm inner diameter (https://www.sigmaaldrich.com/US/en/product/aldrich/z565695 accessed on 17 September 2022). Furthermore, there appears to be a lack of heating mantles smaller than 250 mL capacity (https://chemglass.com/mantles-heating-tops-glas-col accessed on 17 September 2022), miniaturized power sources for these, or very small chiller/circulators. Thus, miniaturization of the apparatuses to a desk-top size is feasible but unlikely to decrease the costs of fabrication or operation significantly.

An alternative is to explore hand-sized microfluidic types of apparatuses printed in glass or other chemical and heat resistant materials. Such devices would be reproducible in multiple copies at significantly lower costs than the current apparatuses and therefore easily contained in a chemical hood for safety when using gases such as ammonia, hydrogen sulfide, carbon monoxide, etc. Another advantage would be that microfluidic devices can be designed to operate at very high gas pressures, such as might be found on gas giant planets, and at very low temperatures such as are usually found there. Such apparatuses might integrate all of the selection factors of ReBioGeneSys 2.0 or separate devices could be designed, each to carry out one of the selection processes, as in ReBioGeneSys 1.0. Separate devices could then be ganged together in whatever order desired. Yet, another possibility would be to have a single, very minimalistic design that could be exposed to each selection factor independently, e.g., to UV light/dark cycles delivered by a separate UV light source and then to freeze/thaw cycles implemented by literally placing the apparatus in a freezer and taking it out. One additional advantage of such microfluidic apparatuses would be the ease with which they might be directly integrated into HPLC or GC/Mass Spec analytical equipment for continuous monitoring of products while obviating risk of contamination that might occur from human sampling of the apparatus. The use of multiple, miniaturized apparatuses would also make it much easier to run side-by-side experiments comparing the effects of different selection factors on a common set of starting reactants, or the effects of different combinations of such selection factors or their order.

These apparatuses, whether large or miniaturized, may also permit experimental exploration of strategies by which nature controlled the combinatorial chemical explosion problem that must accompany experiments utilizing complex mixtures of compounds (so-called “dirty experiments”) over long periods of times. While it is logical to assume that the longer a chemical experiment is run, the worse the combinatorial chemical explosion problem will become [22], we predict that the addition of selection processes will mitigate the problem. Indeed, Cronin et al. [13] have found that it is possible to tame the combinatorial explosion of the formose reaction when run over extended periods of time by “seeding” the product mixture into a fresh version of the reaction and by using various mineral surfaces. They observed that the overall number of products decreased as the number of cycles increased, suggesting that as more complex molecules evolve, they begin to compete for available chemical precursors, thereby restraining the combinatorial explosion. Further experiments of this type augmented by cycles of selection may be similarly revealing.

Another possible outcome of implementing selection pressures may result in another outcome that can control the combinatorial chemical explosion problem and that is to evolve metabolic replication cycles. Baum and his colleagues have demonstrated that chemical ecosystem selection can be performed by repeatedly seeding a synthetic “prebiotic soup” onto pyrite grains to yield mutually reinforcing sets of catalyzed reactions [12]. Evolving such autocatalytic sets of reactions from mutually reinforcing sets or reactions are a clear desiderata for prebiotic chemical studies, since all known biological systems implement such cycles [88].

Finally, implementing selection pressures may also yield the third of limiting a combinatorial chemical explosion, which is through the evolution of functional modules through molecular complementarity [89]. Molecular complementarity may be observable in small-molecule interactions that protect the constituents of the complex against degradation processes thereby increasing the concentrations of the participating compounds compared with other, non-complementary molecules; it may foster the emergence of peptide or ribonucleotide catalysts; and polymerization reactions driven by freeze–thaw and wet/dry reactions (see Section 1.1 and Section 1.2) may yield self-organizing molecules (phospholipids, polysaccharides, polypeptide aggregates, or polyribonucleosides) as well as self-replicating polymers. Regarding these possibilities, it is important to note that some amino acids, have been demonstrated to catalyze many types of chemical reactions (e.g., [90,91], many of relevance to prebiotic chemical reactions (e.g., synthesis of sugars [92]); that some peptides catalyze specific chemical reactions [93]; while other peptides, such as poly-serines, are capable of self-replication reactions [94]. Thus, future studies should look not only for the emergence of polyribonucleosides and their autocatalytic sets [94,95,96], but also for the emergence small-molecule catalyzed reactions more broadly and synergies between nucleic acids, amino acids, lipids, etc. [97].

In sum, new apparatuses have been designed to explore the roles of selection processes that may have been at work in various prebiotic chemical ecosystems, either alone or in combination, and to make it possible to determine whether prebiotic natural selection leads to increased synthesis of polymerized products, emergence of self-organization and replication of selected sets of products, and control of the combinatorial chemical explosion that would be expected in the absence of selection. These apparatuses are, hopefully, just the first of many, improved versions that foster novel experimentation in prebiotic chemical ecosystems.

## Figures and Tables

**Figure 1 life-12-01508-f001:**
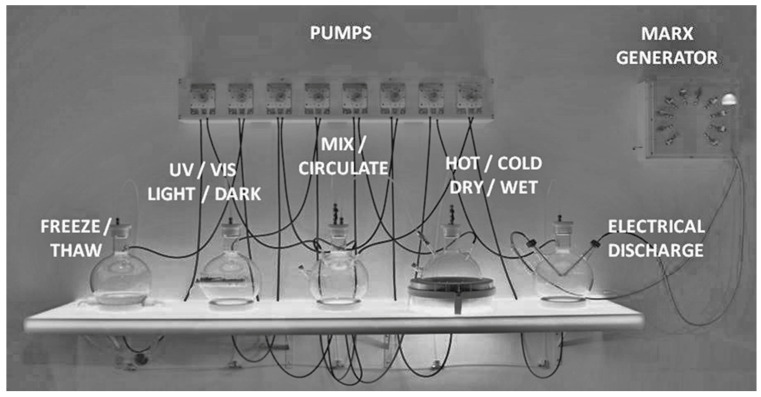
Overview of the main elements of ReBioGeneSys showing the peristaltic pump array at the top that circulates the fluid from one flask to another in any order desired and, from left to right, the flask through which the chiller fluid circulates antifreeze fluid (left), which permits freezing/thawing depending on the temperature setting of the chiller; a flask modified to permit insertion of an ultraviolet light or visible light source; a flask (center) through which all the other flasks can circulate their fluid to enable passage to any other flask; a flask sitting in a heating mantle permitting boiling or evaporation of the fluid; and (right) a flask fitted with electrodes connected to the Marx generator that creates electrical discharges mimicking lightning.

**Figure 2 life-12-01508-f002:**
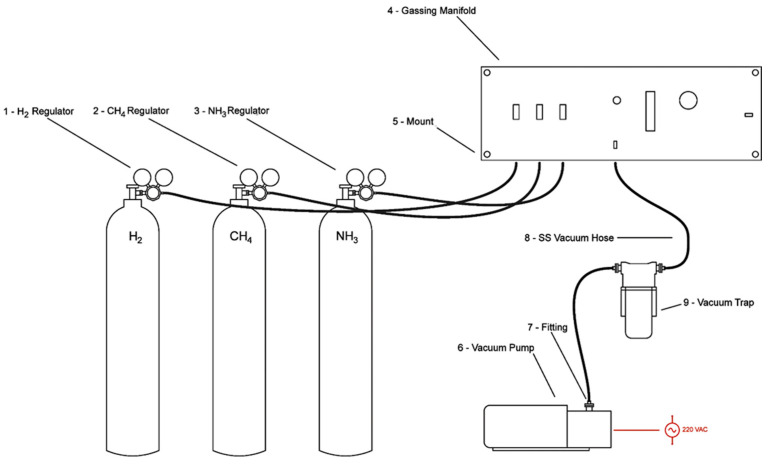
Diagram illustrating how the gas/vacuum manifold integrates gas flow from the gas cylinders into the manifold and into the rest of the apparatus, each controlled by a Swagelok valve, and exiting the manifold through a single Tygon tube to the right (not shown). The valve to the vacuum pump (lower right) is closed during gassing of the apparatus. If the three valves to the gasses are all closed, the valve to the vacuum pump can be opened and the apparatus evacuated. The vacuum pump attaches to the manifold by means of a pair of stainless steel low-pressure vacuum hoses that integrate a Visitrap water and ammonia trap to protect the pump.

**Figure 3 life-12-01508-f003:**
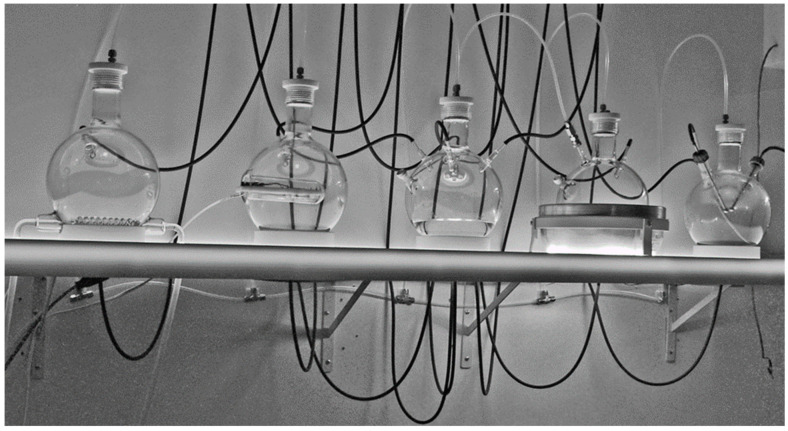
Close up of the flasks showing the two ways in which they are interconnected. The black Tygon tubing connects all of the flasks through the center one via the programmable peristaltic pumping apparatus (not shown). Fluid enters at the upper sides of the flasks and is drained through connectors at the bottom (not visible). The white PFA tubing that connects through the white PFA plugs at the top of each flask connect all of the flasks to the manifold controlling gas entry to the system. All of the flasks are share the same gas environment as there are no valves or pumps in the gas tubing connecting the flasks. The freezing flask has a coiled borosilicate tube inserted through which antifreeze is circulated by a chiller (not shown), which permits the temperature of the flask to be maintained at whatever temperature is desires, including below the freezing point. The UV light flask has an insert into which a light source can be placed. A better version is illustrated in the integrated version, ReBioGeneSys 2.0, illustrated below.

**Figure 4 life-12-01508-f004:**
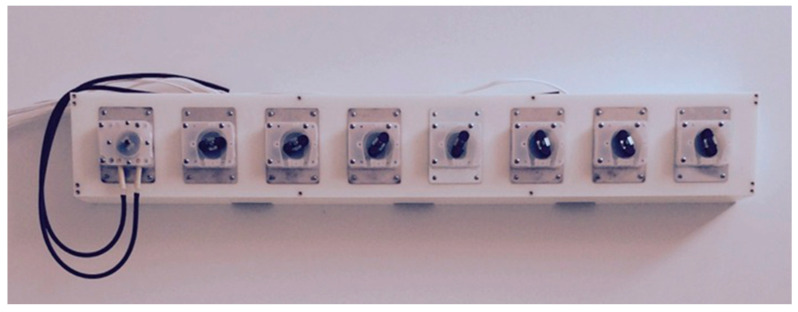
Peristaltic pump array permitting every flask to be connected to the central receiving flask from which every other flask can be reached. The Viton tubing is shown connected only to the left-most pump and the wiring connecting the pump array to the custom programmable electronic control box is visible at the extreme left.

**Figure 5 life-12-01508-f005:**
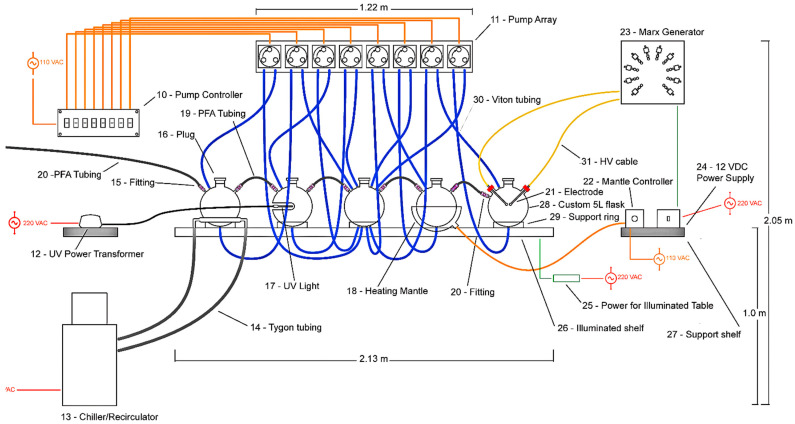
Schematic diagram showing the arrangement of the pump apparatus and its tube connections to the flasks. Each flask has an inlet near its top to allow liquid to be pumped in and an outlet at the bottom to allow liquid to be pumped out. Inlets and outlets are controlled by separate peristaltic pumps (top) controlled by a series of programmable switches (upper left). When a pump is inactive, no liquid can flow in or out of a flask.

**Figure 6 life-12-01508-f006:**
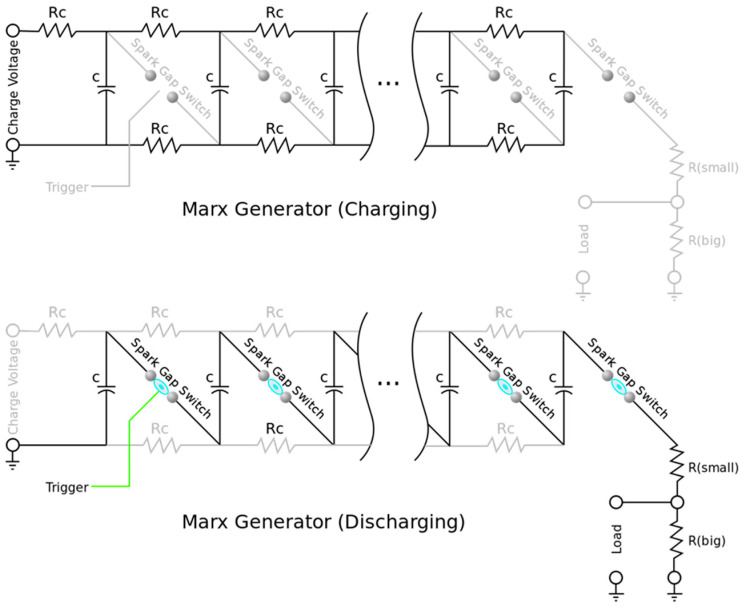
Schematic diagram of a Marx generator, a high voltage circuit used in insulation testing and scientific research. It generates a pulse of high voltage by charging multiple capacitors in parallel and then suddenly connecting them together in series by spark gaps. The capacitors are charged by the resistor network. Although the left capacitor has the greatest charge rate, the generator is typically allowed to charge for a long period of time, and all capacitors eventually reach the same charge voltage. (Wikimedia Commons, https://en.wikipedia.org/wiki/File:Marx_Generator.svg Accessed 28 August 2022).

**Figure 7 life-12-01508-f007:**
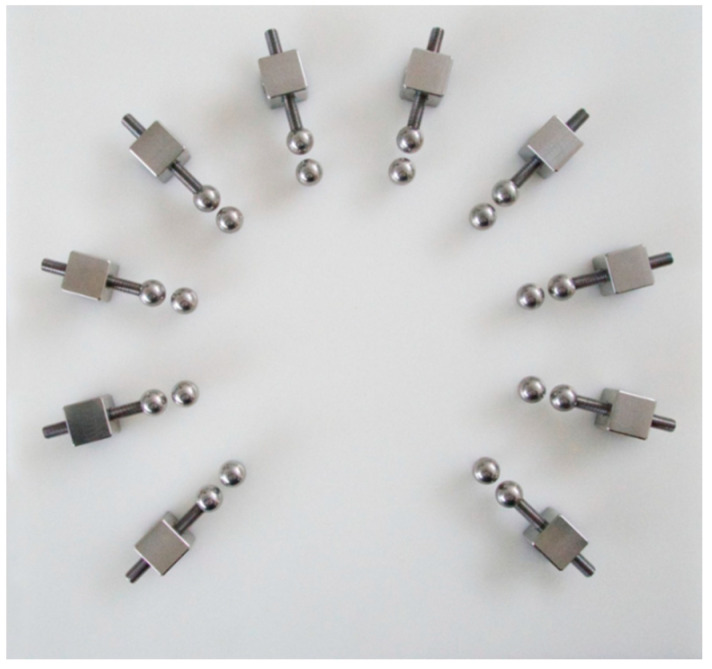
Arrangement of the custom Marx generator showing the spark gap electrodes, which permit the spark gap distance to be varied by means of a threaded screw arrangement.

**Figure 8 life-12-01508-f008:**
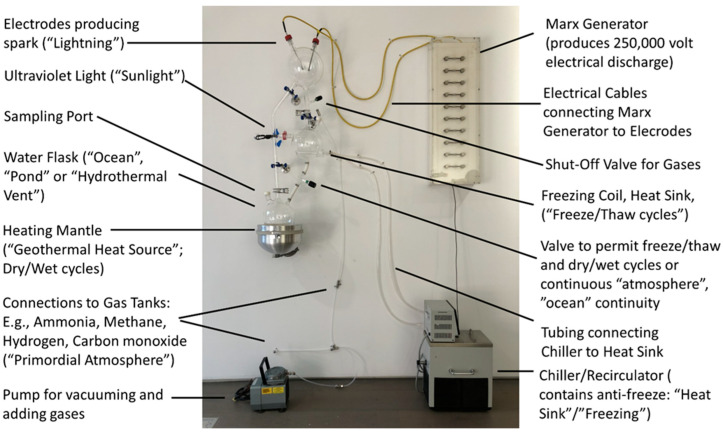
Overview of ReBioGeneSys version 2.0, in which the elements from ReBioGeneSys 1.0 have been reintegrated into a simpler form.

**Figure 9 life-12-01508-f009:**
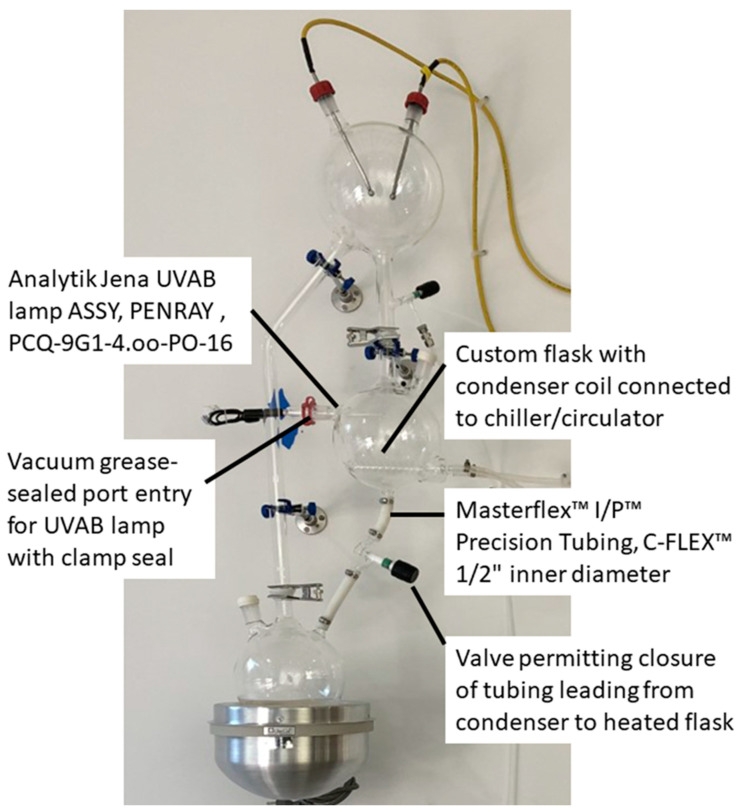
Close-up view of ReBioGeneSys 2.0 highlighting some of the key innovations. A single flask (center) now performs both freeze/thaw and UV light/dark functions, either simultaneously of separately. To use the flask for freezing, water from the heating flask (lower left) can be evaporated into the freezing flask by closing the valve below the freezing flask (the flow of anti-freeze at a temperature below 0 °C acts as a heat sink to balance the heat source evaporating the water in the heating flask). In order to accommodate freezing of the water in the tube connecting the condenser flask to the valve, and to accommodate the need to resist vacuum prior to gassing and re-gassing, and to resist the ammonia in the experimental atmosphere, special Masterflex tubing is required (as noted in the image). Unlike ReBioGeneSys 1.0, the UVAB lamp is inserted directly into the flask via a clamped, vacuum grease-sealed port rather than being set into an indentation in the flask.

**Table 1 life-12-01508-t001:** List of parts for ReBioGeneSys 1.0 keyed to Figures below.

Item Number	Part Description	Quantity
1	High pressure regulator for hydrogen	1
2	High pressure regulator for methane	1
3	Anhydrous ammonia regulator	1
4	Custom aluminum and stainless steel gassing manifold	1
5	Aluminum standoff mounts for the gassing manifold	4
6	Adixen (Pfeiffer) Chemical Pascal 2005C1 3.5CFM vacuum	1
7	Stainless steel KF-25 to ¼” compression fitting, centering ring and clamp	3
8	Stainless steel braided low pressure vacuum hose	2
9	Visitrap water and ammonia vacuum trap	1
10	Custom switching controller for peristaltic pump array	1
11	Custom peristaltic pump array made of Delrin	1
12	Ultraviolet light power transformer	1
13	Cole Parmer recirculating heater/chiller	1
14	½” OD Tygon tubing	2
15	¼” stainless steel Swagelok Ultra Torr to compression fitting: SS-4-UT-6-400	1
16	50 mm PFA threaded plug	5
17	Analytic Jena UV(AB) light	1
18	Eating mantle for 5 L round bottom flask	1
19	3/8” PFA tubing	4
20	3/8” stainless steel Swagelok Ultra Torr to compression fitting: SS-6-UT-6-600	8
21	Custom stainless steel electrodes with round ball caps	2
22	Heating mantle controller	1
23	Custom Marx generator, HV electronics, stainless steel and Delrin	1
24	12 VDC variable power transformer for Marx generator	1
25	Power for illuminated shelf	1
26	Custom made Corian shelf with LED illumination	1
27	Corian support shelf made to match main Corian shelf	1
28	5L custom borosilicate glass round bottom flask, modified to accept input and output tubing, threaded plugs, UV light, chiller condenser, heater, or electrodes	5
29	Support ring	4
30	Masterflex FDA Viton tubing L/S #96412-25	16
31	Heavy insulated high voltage cable	2

## Data Availability

There is no data associated with this paper.

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
