# Peer review of "Novel Apparatuses for Incorporating Natural Selection Processes into Origins-of-Life Experiments to Produce Adaptively Evolving Chemical Ecosystems"

_life, 2022, doi:10.3390/life12101508_

Round 1

Reviewer 1 Report

Title: Novel Apparatuses for Incorporating Natural Selection Processes into Origins-of-Life Experiments to Produce Adaptively Evolving Chemical Ecosystems

Overview and general recommendation:

Prebiotic chemistry has moved from simple experiments to more complex ones, the last experiments intend to include multiple variables and reproduce “natural environments” that are clearly complicated. In general, this is reached in multiple steps experimental design. The article describes the design of complex apparatuses to achieve the multi-variable approach that prebiotic chemistry needs. The authors firstly describe the relevance of this kind of studies in the field and numbered some typical experiments in which one or more variable are studied. Then, they described the apparatuses they built. This paper is very relevant in the field, and it shows a novel design for prebiotic chemistry devices. I just want to mention some points before publication.

Major comments

Is it correct that the Simple Summary and the Abstract are almost the same?

Some figures need improvement, particularly the ones with embedded small text, which are illegible.

Authors mentioned the expression “chemical ecology” I’m not familiar with that term, so it could be useful to explain the meaning of this expression.

Minor comments

Along the text I made some observations, please check it.

Author Response

REVIEWER 1

Open Review

(x) I would not like to sign my review report

( ) I would like to sign my review report

English language and style

( ) Extensive editing of English language and style required

( ) Moderate English changes required

(x) English language and style are fine/minor spell check required

( ) I don't feel qualified to judge about the English language and style

                Yes         Can be improved              Must be improved           Not applicable

Does the introduction provide sufficient background and include all relevant references?

                (x)           ( )            ( )            ( )

Are all the cited references relevant to the research?

                (x)           ( )            ( )            ( )

Is the research design appropriate?

                (x)           ( )            ( )            ( )

Are the methods adequately described?

                (x)           ( )            ( )            ( )

Are the results clearly presented?

                ( )            (x)           ( )            ( )

Are the conclusions supported by the results?

                ( )            (x)           ( )            ( )

Comments and Suggestions for Authors

Title: Novel Apparatuses for Incorporating Natural Selection Processes into Origins-of-Life Experiments to Produce Adaptively Evolving Chemical Ecosystems

Overview and general recommendation:

Prebiotic chemistry has moved from simple experiments to more complex ones, the last experiments intend to include multiple variables and reproduce “natural environments” that are clearly complicated. In general, this is reached in multiple steps experimental design. The article describes the design of complex apparatuses to achieve the multi-variable approach that prebiotic chemistry needs. The authors firstly describe the relevance of this kind of studies in the field and numbered some typical experiments in which one or more variable are studied. Then, they described the apparatuses they built. This paper is very relevant in the field, and it shows a novel design for prebiotic chemistry devices. I just want to mention some points before publication.

Major comments

Is it correct that the Simple Summary and the Abstract are almost the same?

Yes. We have rewritten the simple summary.

Some figures need improvement, particularly the ones with embedded small text, which are illegible.

Yes! We also noticed that some did not reproduce well and have prepared and inserted ones with larger fonts and lines.

Authors mentioned the expression “chemical ecology” I’m not familiar with that term, so it could be useful to explain the meaning of this expression.

We have removed the term: it is not necessary, especially if it causes confusion!

Minor comments

Along the text I made some observations, please check it.

Thank you! Done!

Reviewer 2 Report

This paper describes two newly developed set-ups for conducting prebiotic synthesis in conditions that mimic the environmental heterogeneity of the early Earth. The effort is justified by the claim that long-term experiments that incorporate many environmental factors are most likely to detect the earliest stages of abiogenesis, manifested perhaps as the taming of chemical complexity.

My overall assessment is that these new apparatuses are potentially powerful new tools for probing the emergence of life. I could not agree more that we should be conducting experiments that run for a long time and combine being open, in the sense of featuring a flux of replenishing food, with selection, which is to say conditions that might favor some autocatalytic chemical systems over others. The apparatuses are cool, and I know I would love to be able to deploy them in my lab! I am very supportive of publication, but I would like to see a somewhat more sophisticated discussion of the underlying evolutionary principles and more clarity on results a successful experiment would yield.

I know this sounds horribly self-serving, but I do feel that you should look at two of my prior papers. My 2016 paper with Kalin Vetsigian (https://doi.org/10.1007/s11084-016-9526-x) directly discusses the question of how we can (and should) deploy in vitro selection on chemical ecosystems (a term that I think we coined) to study the origin of life. You might also consult our 2019 paper where we implemented chemical ecosystem selection using a serial, transfer-and-dilution protocol (https://doi.org/10.3390/life9040080). Our theoretical work (e.g., https://doi.org/10.1371/journal.pcbi.1010498) may also be of interest.

Below I list some specific requests and suggestions.

1)    The two abstracts are identical, which is a waste of space.

2)    You imply that the goal of your experimental research is to observe “chemical systems to evolve into biological ones.” This begs the question of what makes a system chemical or biological. Since I think that the difference comes down to the ability to complexify via adaptive evolution, would you say that your goal is to look for the spontaneous emergence of adaptive evolution?

3)    You state that “Evolution by natural selection has four requirements.” This formulation is okay as a first approximation, but a bit naïve and needing some caveats. I would suggest reading Peter Godfrey-Smith’s book Darwinian Populations and Natural Selection and other papers that have thought about possibilities of adaptive evolution prior to the appearance of genetic polymers.

4)    “We argue that it is time to begin experimenting with the long-term effects of the prebiotic natural selection processes that may have aided biotic life to emerge by taming the combinatorial chemical explosion.” Couldn’t agree more! Maybe say this in the abstract?

5)    “a constant resupply of chemical reactants was available to drive production of products (“species reproducibility”).” I agree with the need to maintain an open (driven) ecosystem. However, I don’t understand the parenthetical clause.

6)    The review of “processes” in Sects. 1.1-1.6 seems somewhat overly detailed to me.

7)    There are probably a few more photos than are really needed for the paper. Extras can go in supplemental information. Indeed the results text seems rather repetitive and much of it could go into supp info.

8)    A major gap in the paper, in my opinion, is a lack of discussion of what features would be looked for. The paper implies that products could be tracked over time using GC-MS, but does not include any clear statement about what would make a result more or less interesting. Would you be looking for particular compounds? Or would you be looking for dynamical patterns indicative of autocatalysis and adaptive evolution? And if the latter, what are these patterns?

9)    The apparatuses are cool, but they are large and could not easily be replicated at scale. You could add a section in the discussion on whether miniaturization is possible or even microfluidics.

Author Response

Reviewer 2

Open Review

( ) I would not like to sign my review report

(x) I would like to sign my review report

English language and style

( ) Extensive editing of English language and style required

( ) Moderate English changes required

(x) English language and style are fine/minor spell check required

( ) I don't feel qualified to judge about the English language and style

                Yes         Can be improved              Must be improved           Not applicable

Does the introduction provide sufficient background and include all relevant references?

                ( )            (x)           ( )            ( )

Are all the cited references relevant to the research?

                (x)           ( )            ( )            ( )

Is the research design appropriate?

                (x)           ( )            ( )            ( )

Are the methods adequately described?

                (x)           ( )            ( )            ( )

Are the results clearly presented?

                (x)           ( )            ( )            ( )

Are the conclusions supported by the results?

                ( )            ( )            ( )            (x)

Comments and Suggestions for Authors

This paper describes two newly developed set-ups for conducting prebiotic synthesis in conditions that mimic the environmental heterogeneity of the early Earth. The effort is justified by the claim that long-term experiments that incorporate many environmental factors are most likely to detect the earliest stages of abiogenesis, manifested perhaps as the taming of chemical complexity.

My overall assessment is that these new apparatuses are potentially powerful new tools for probing the emergence of life. I could not agree more that we should be conducting experiments that run for a long time and combine being open, in the sense of featuring a flux of replenishing food, with selection, which is to say conditions that might favor some autocatalytic chemical systems over others. The apparatuses are cool, and I know I would love to be able to deploy them in my lab! I am very supportive of publication, but I would like to see a somewhat more sophisticated discussion of the underlying evolutionary principles and more clarity on results a successful experiment would yield.

We have provided additional information as requested.

I know this sounds horribly self-serving, but I do feel that you should look at two of my prior papers. My 2016 paper with Kalin Vetsigian (https://doi.org/10.1007/s11084-016-9526-x) directly discusses the question of how we can (and should) deploy in vitro selection on chemical ecosystems (a term that I think we coined) to study the origin of life. You might also consult our 2019 paper where we implemented chemical ecosystem selection using a serial, transfer-and-dilution protocol (https://doi.org/10.3390/life9040080). Our theoretical work (e.g., https://doi.org/10.1371/journal.pcbi.1010498) may also be of interest.

No need to apologize. If any apology is needed, it is from us for overlooking these insightful papers. Thank you for the references. We have added these and provided discussions of their contents in the  Introduction and Discussion where appropriate.

 Below I list some specific requests and suggestions.

1)    The two abstracts are identical, which is a waste of space.

Agree! We have substantially rewritten the Simple Abstract.

2)    You imply that the goal of your experimental research is to observe “chemical systems to evolve into biological ones.” This begs the question of what makes a system chemical or biological. Since I think that the difference comes down to the ability to complexify via adaptive evolution, would you say that your goal is to look for the spontaneous emergence of adaptive evolution?

Thank you for pointing out the difficulties with our statement. We have decided to remove the phrase “into biological ones”. That phrase is, indeed, difficult or impossible to define in practice. We have clarified the point to say that we hope to obtain the “spontaneous emergence of adaptive chemical ecosystems” as defined by the criteria in Introduction as having the properties of “reproduction”, variation, non-random selection, and adaptation.

3)    You state that “Evolution by natural selection has four requirements.” This formulation is okay as a first approximation, but a bit naïve and needing some caveats. I would suggest reading Peter Godfrey-Smith’s book Darwinian Populations and Natural Selection and other papers that have thought about possibilities of adaptive evolution prior to the appearance of genetic polymers.

Thank you for this suggestion. You’re right: our language was sloppy. We have modified our formulation in light of PGS’s clarifications.

4)    “We argue that it is time to begin experimenting with the long-term effects of the prebiotic natural selection processes that may have aided biotic life to emerge by taming the combinatorial chemical explosion.” Couldn’t agree more! Maybe say this in the abstract?

Excellent idea: abstract modified!

5)    “a constant resupply of chemical reactants was available to drive production of products (“species reproducibility”).” I agree with the need to maintain an open (driven) ecosystem. However, I don’t understand the parenthetical clause.

The parenthetical phrase refers back to the need for evolving species to “reproduce”, which was one of our criteria for defining/identifying an adaptive system. Since it appears to be a distraction to refer back to this here, we have removed the parenthetical phrase and, instead, used your term of an open (driven) ecosystem.

6)    The review of “processes” in Sects. 1.1-1.6 seems somewhat overly detailed to me.

Perhaps, but as a result of reviews of some of our other papers, we suspect that very few people are aware of the range of methods used to augment prebiotic chemical processes or their application to amino acid/peptide, nucleic acid, sugar and lipid syntheses; even fewer seem to be aware of their limited use for selection.  We’d rather assume too little and explain too much than leave some readers confused, and we’d rather cite too many studies and be inclusive than cite to few and get someone’s back up. Hope that’s okay with you….

7)    There are probably a few more photos than are really needed for the paper. Extras can go in supplemental information. Indeed the results text seems rather repetitive and much of it could go into supp info.

We’ve edited this to delete some of the redundancy.

8)    A major gap in the paper, in my opinion, is a lack of discussion of what features would be looked for. The paper implies that products could be tracked over time using GC-MS, but does not include any clear statement about what would make a result more or less interesting. Would you be looking for particular compounds? Or would you be looking for dynamical patterns indicative of autocatalysis and adaptive evolution? And if the latter, what are these patterns?

We have tried to clarify these points in our revised Discussion. Your papers are very valuable in this regard, thanks!

9)    The apparatuses are cool, but they are large and could not easily be replicated at scale. You could add a section in the discussion on whether miniaturization is possible or even microfluidics.

Good points. We have been considering both possibilities and now provide a paragraph discussing these and some of the construction challenges that they pose as well as possible benefits.